# Utility of CT to Differentiate Pancreatic Parenchymal Metastasis from Pancreatic Ductal Adenocarcinoma

**DOI:** 10.3390/cancers13133103

**Published:** 2021-06-22

**Authors:** Maxime Barat, Rauda Aldhaheri, Anthony Dohan, David Fuks, Alice Kedra, Christine Hoeffel, Ammar Oudjit, Romain Coriat, Maximilien Barret, Benoit Terris, Ugo Marchese, Philippe Soyer

**Affiliations:** 1Department of Radiology, Hôpital Cochin, AP-HP, 75014 Paris, France; dr.raudaa@gmail.com (R.A.); anthony.dohan@aphp.fr (A.D.); alice.kedra@aphp.fr (A.K.); ammar.oudjit@aphp.fr (A.O.); philippe.soyer@aphp.fr (P.S.); 2Faculté de Médecine, Université de Paris, 75006 Paris, France; david.fuks@aphp.fr (D.F.); romain.coriat@aphp.fr (R.C.); maximilien.barret@aphp.fr (M.B.); benoit.terris@aphp.fr (B.T.); ugo.marchese@aphp.fr (U.M.); 3Department of Digestive, Hepatobiliary and Endocrine Surgery, Hôpital Cochin, AP-HP, 75014 Paris, France; 4Department of Radiology, Hôpital Robert Debré, 51092 Reims, France; choeffel-fornes@chu-reims.fr; 5Department of Gastroenterology, Hôpital Cochin, AP-HP, 75014 Paris, France; 6Department of Pathology, Hôpital Cochin, AP-HP, 75014 Paris, France

**Keywords:** carcinoma, pancreatic ductal, pancreatic neoplasms, tomography, X-ray computed

## Abstract

**Simple Summary:**

The purpose of this retrospective study was to report the computed tomography (CT) features of pancreatic parenchymal metastasis (PPM) and identify CT features that may help discriminate between PPM and PDAC. At multivariable analysis, well-defined margins (OR, 6.64; 95% CI: 1.47–29.93; *p* = 0.014), maximal enhancement during arterial phase (OR, 6.15; 95% CI: 1.13–33.51; *p* = 0.036), no vessel involvement (OR, 7.19; 95% CI: 1.51–34.14) and no Wirsung duct dilatation (OR, 10.63; 95% CI: 2.27–49.91) were independently associated with PPM. A nomogram based on CT features identified at multivariable analysis yielded an AUC of 0.92 (95% CI: 0.85–0.98) for the diagnosis of PPM vs. PDAC.

**Abstract:**

**Purpose**: To report the computed tomography (CT) features of pancreatic parenchymal metastasis (PPM) and identify CT features that may help discriminate between PPM and pancreatic ductal adenocarcinoma (PDAC). **Materials and methods**: Thirty-four patients (24 men, 12 women; mean age, 63.3 ± 10.2 [SD] years) with CT and histopathologically proven PPM were analyzed by two independent readers and compared to 34 patients with PDAC. Diagnosis performances of each variable for the diagnosis of PPM against PDAC were calculated. Univariable and multivariable analyses were performed. A nomogram was developed to diagnose PPM against PDAC. **Results**: PPM mostly presented as single (34/34; 100%), enhancing (34/34; 100%), solid (27/34; 79%) pancreatic lesion without visible associated lymph nodes (24/34; 71%) and no Wirsung duct enlargement (29/34; 85%). At multivariable analysis, well-defined margins (OR, 6.64; 95% CI: 1.47–29.93; *p* = 0.014), maximal enhancement during arterial phase (OR, 6.15; 95% CI: 1.13–33.51; *p* = 0.036), no vessel involvement (OR, 7.19; 95% CI: 1.512–34.14) and no Wirsung duct dilatation (OR, 10.63; 95% CI: 2.27–49.91) were independently associated with PPM. The nomogram yielded an AUC of 0.92 (95% CI: 0.85–0.98) for the diagnosis of PPM vs. PDAC. **Conclusion**: CT findings may help discriminate between PPM and PDAC.

## 1. Introduction

Pancreatic ductal adenocarcinoma (PDAC) represents 90% of all pancreatic malignant tumors [1]. However, myriad tumors can develop in the pancreas and thus one major role of imaging is lesion characterization [2,3,4,5,6]. Owing to high capabilities for tissue characterization, magnetic resonance imaging has demonstrated utility for the diagnosis of pancreatic lesions [7,8]. However, computed tomography (CT) remains the first line imaging modality for the diagnosis of pancreatic tumors [2,3,4,5,6].

Pancreatic parenchyma metastasis (PPM) is a rare tumor that represents 2–5% of all malignant tumors of the pancreas [9]. PPM is a rare condition that is found in 3–12% of patients with advanced cancer in autopsy studies [9,10,11] and the time interval between the diagnosis of primary cancer and that of PPM usually ranges between one and three years, although longer time intervals have been reported [11]. The most frequent primary cancers that give PPM are renal cancers, breast cancers, lung cancers, colorectal cancers and skin melanomas [12,13,14,15,16]. The specific diagnosis of PPM is often delayed because patients with this condition may present with nonspecific symptoms or even with no symptoms at all in 54.5% to 83% of patients [17,18,19] and PPM are often detected incidentally on CT performed for clinical surveillance or follow-up [20]. In addition, results from autopsy series reveal that up to 30% of PPMs are clinically erroneously considered as primary pancreatic cancers [21]. As a result, imaging has a major role in the detection of PPM in patients with a known cancer [20,22]. This role is rendered more critical because depiction and characterization of PPM may alter the current treatment received by the patient and may indicate surgical resection in specific situations [9,16,18,19,23,24].

The CT features of PPM have been reported in some articles and researchers have suggested that PPM may display orientating features that mirror those of the primary cancer [17,20,22]. However, PPM has a broad spectrum of CT presentations and may have nonspecific features making the diagnosis difficult on CT [25,26]. In addition, in patients with prior malignancy, a newly developed pancreatic mass is due to PPM in only 40% of them, so that the diagnosis of PPM is not so straightforward using clinical history [15]. To our knowledge, no studies have directly compared the CT features of PPM to those of PDAC.

This study was designed to report the CT features of PPM and identify imaging characteristics that help discriminate between PPM and PDAC.

## 2. Materials and Methods

### 2.1. Patients

This retrospective study was approved by institutional review board (AAA-2021-08019) and requirement for written informed consent was waived.

The database of the department of pathology of our institution was queried from January 2005 to December 2020. The initial search retrieved 137 patients with possible PPM. Two patients were initially excluded because they were originally misclassified. A cross-match was performed with the radiology department database to identify those who had undergone CT examination in our institution. One hundred and one patients were further excluded because they were referred from another institution for endoscopic biopsy and did not undergo CT in our institution (*n* = 60) or because no CT examination was available for review (*n* = 41). The patient inclusion process is summarized in Figure 1.

The final study population consisted of 34 patients with PPM. There were 22 men and 12 women (mean age, 66.5 ± 10.7 [standard deviation (SD)] years; range: 40–82 years) (Table 1).

PPM was confirmed after histopathological analysis of biopsy specimens or after surgical resection of PPM. The primary cancers were renal cell carcinoma (RCC) (*n* = 11 patients), skin melanoma (*n* = 4 patients), Merkel cell carcinoma (*n* = 3 patients), lung cancer (squamous cell carcinoma, *n* = 4 patients; adenocarcinoma, *n* = 3 patients), adrenocortical carcinoma (*n* = 3 patients), colorectal adenocarcinoma (*n* = 3 patients), gastric adenocarcinoma (*n* = 1 patient), vesicular thyroid carcinoma (*n* = 1 patient) and bladder urothelial carcinoma (*n* = 1 patient). All patients had a known primary cancer at the time of diagnosis of PPM and four patients had two coexisting cancers as possible cause of PPM. Seventeen patients (17/34; 50%) had isolated PPM with no local recurrence of the primary cancer and no other metastatic sites.

Thirty-four patients with PDAC were identified for matched comparison. They were extracted from a database of 367 patients with histopathologically proven PDAC. These 34 patients were thus selected according to sex, age, tumor largest diameter and underwent CT examination during the same time frame and with the same CT protocols than those with PPM. There were 22 men and 12 women, with a mean age of 63.6 ± 8.9 (SD) years (range: 40–82 years). PDAC was confirmed after histopathological analysis of biopsy specimens obtained during surgery (13/34; 38%) or endoscopic biopsy (21/34; 62%). In this group, 3/34 patients (9%) had Stage IA PDAC, 5/34 (15%) had Stage IB PDAC, 3/34 (9%) had Stage IIA PDAC, 9/34 (26%) had Stage IIB PDAC, 6/34 (18%) had Stage III PDAC and 8/34 (23%) had Stage IV PDAC according to the American Joint Committee on Cancer (Appendix A) [27]. Four patients had prior history of another cancer including breast carcinoma, lung carcinoma, colon adenocarcinoma and skin melanoma (one patient each).

### 2.2. CT Protocol

CT examinations were performed with a single source helical CT equipment (Revolution HD^®^, General-Electric Healthcare, Wauwatosa, WI, USA; Somatom Sensation^®^ 64, Siemens Healthineers; or Somatom Definition^®^ Flash, Siemens Healthineers, Erlangen, Germany). Acquisition parameters were as follows: field-of-view, 279–350 mm; beam collimation, 38.4–40 mm (64 × 0.6–0.625 mm collimator setting); slice thickness, 1–1.25 mm; peak tube potential, 110–120 kVp; gantry revolution time, 0.5 s; and beam pitch, 0.984–1.2.

Iodinated contrast material (iomeprol, Iomeron 350^®^, Bracco Imaging; or iobitridol, Xenetix 350^®^, Guerbet, Aulnay-sous-Bois, France) was injected intravenously with an automated power injector (rate, 2.5–4 mL/s; total volume, 95–125 mL). After unenhanced acquisition, arterial phase (35–45 s after initiating contrast material administration) and portal venous phase (delay, 65–80 s) acquisitions were obtained.

### 2.3. Image Analysis

Two radiologists with 5-(R.A.) and 33-(P.S.) years of experience in pancreatic imaging independently reviewed the CT examinations on a picture archiving and communication system viewing station (Directview, 12.1.0365 version, Carestream Health Inc., Rochester, NY, USA). Anonymized CT examinations were analyzed with the radiologists blinded to any patient information. At the end of the independent readings, a consensus reading was performed to obtain a consensus opinion for qualitative variables for further statistical analysis.

CT images were analyzed by using a standardized data collection form (Appendix B). PPMs and PDACs were evaluated for the following features: largest axial diameter, location (head, body or tail), shape (oval or round), margin (well or ill-defined contours), enhancing rim/tumor capsule, content (solid, cystic or mixed), internal necrosis, internal calcification, homogeneity of tumor enhancement after intravenous administration of iodinated contrast material, degree of tumor enhancement relative to the apparently uninvolved pancreas on arterial and portal phases, and imaging phase with maximal enhancement. The largest axial diameter was measured using calipers in the axial plane on magnified CT images. Non-enhancing areas with attenuation similar to that of the gallbladder on unenhanced CT images were considered as necrotic components of the tumor, whereas the others were considered solid [6]. Tumor enhancement was considered present when enhancement was identified on CT images obtained after intravenous administration of iodinated contrast material, whatever the specific imaging phase. Calcifications were searched on unenhanced images. Vascular involvement was considered for involvement (encasement/occlusion) of any vascular structure (arterial or venous) identified on enhanced CT images.

Other features included: presence of Wirsung duct dilatation (diameter >4 mm), upstream pancreatic atrophy, segmental portal hypertension, hepatic metastases, bile duct dilatation, presence of visible lymph nodes, largest axial diameter of visible lymph nodes, direct involvement of adjacent organ, mesenteric panniculitis [28] and ascites.

### 2.4. Statistical Analysis

Statistical analysis was performed using software (SAS, V 9.3, SAS Institute; R-3.5.3, R Project, R Foundation, Vienna, Austria). Quantitative variables were expressed as means, SD and ranges. Qualitative data were expressed as raw numbers, proportions and percentages. Qualitative variables were compared using Fisher exact test and quantitative variables with Mann–Whitney U test. Agreement between observers for the presence of CT variables was assessed with the Cohen kappa-test using degrees of agreement reported elsewhere [29].

The capabilities of qualitative CT variables for the diagnosis of PPM were evaluated in terms of sensitivity, specificity and accuracy with their corresponding 95% confidence intervals (CIs) using the results of the consensus reading. Qualitative variables were then entered into univariable analysis with a conditional logistic regression model to identify variables associated with PPM at CT. The exact method was used when there was either a complete or a quasi-complete separation of data. Multivariable analysis was performed using a logistic regression model with forward stepwise selection of covariates. Correlations between all variables were searched for. When two variables strongly correlated, only one was included in the multivariable model. All statistical tests were two-sided. A *p* value < 0.05 was considered to indicate significant difference.

A continuous score for PPM was created based on the final multivariable model via a linear combination of selected features that were weighted by their respective coefficients and further presented as a nomogram. The discriminatory capability of the score was evaluated using receiving operative curve (ROC) analysis with calculation of area under the ROC (AUROC).

## 3. Results

### 3.1. Results of Descriptive Statistics

No differences in age and gender distribution were found between patients with PPM and those with PDAC (Table 1). Sensitivity, specificity and accuracy of qualitative variables are reported in Table 2 with corresponding 95% CIs.

Interobserver agreement was substantial to perfect (kappa range: 0.717–1) for all qualitative CT variables of PPM. Detailed kappa values for qualitative CT variables are presented in Appendix C.

The results of descriptive analysis based on the consensus reading are reported in Table 3. PPM presented as a single (34/34; 100%), oval (19/34; 56%) and enhancing (34/34; 100%) pancreatic mass with purely solid content (27/34; 79%) and well-defined margins (23/34; 68%), with a mean largest axial diameter of 35.0 ± 21.10 (SD) mm (Figure 2). Homogeneous distribution was found among pancreatic head, body and tail (*p* = 0.14). Visible lymph nodes were less frequent in PPMs than in PDACs (*p* = 0.002) but, when visible, were larger than those observed in PDACs (*p* = 0.014) (Figure 3). PPMs were more frequently hyperattenuating relative to the apparently uninvolved pancreatic parenchyma on the arterial phase than PDACs ([11/34; 32%] vs. [0/34; 0%], respectively) (*p* < 0.001) and more frequently presented with well-defined tumor margins than PDACs (*p* = 0.005). Maximal tumor enhancement was observed during the arterial phase in 19/34 PPMs (56%) compared to 4/34 PDACs (12%) (*p* < 0.001).

The sensitivity, specificity and accuracy of each CT variable for the diagnosis of PPM are reported in Table 3. Best accuracies for the diagnosis of PPM against PDAC were obtained by absence of Wirsung duct enlargement (76%), maximal tumor enhancement on arterial phase (72%), absence of bile duct dilatation (72%), absence of upstream pancreatic atrophy (71%) and absence of vascular involvement (71%).

Well-defined tumor margins were more frequent in PPM (23/34; 68%) than in PDAC (7/34; 21%) (*p* < 0.001) and this finding yielded 74% accuracy for the diagnosis of PPM against PDAC (Figure 4). Absence of Wirsung duct dilatation and absence of upstream pancreatic atrophy were more frequently observed in PPM (29/34; 85% for both) than in PDAC (11/34; 32% and 15/34; 44%, respectively) (Figure 5) (*p* < 0.001 and *p* = 0.001, respectively) and yielded 76% and 71% accuracies for the diagnosis of PPM. Bile duct dilatation was less frequently observed in PPM (4/34; 12%) than in PDAC (19/34; 56%) (*p* < 0.001). No significant differences were found between PPM and PDAC for all other quantitative variables (Table 3).

### 3.2. Results of Univariable Analysis

The results of univariable analysis are described in Table 4. Well-defined tumor margins (*p* < 0.001), hyperattenuation on arterial phase (*p* < 0.001), maximal tumor enhancement on arterial phase (*p* < 0.001), absence of Wirsung duct enlargement (*p* < 0.001), absence of upstream pancreatic atrophy (*p* < 0.001), absence of vascular involvement (*p* = 0.001) and absence of bile duct dilatation (*p* < 0.001) were the most discriminating CT findings for the diagnosis of PPM vs. PDAC.

### 3.3. Results of Multivariable Analysis and Nomogram

At multivariable analysis, well defined margins (odds ration [OR], 6.64; 95% CI: 1.47–29.93; *p* = 0.014), maximal enhancement during arterial phase (OR, 6.15; 95% CI: 1.13–33.51; *p* = 0.036), absence of vessel involvement (OR, 7.19; 95% CI: 1.51–34.14; *p* = 0.013) and absence of Wirsung duct dilatation (OR, 10.63; 95% CI: 2.27–49.91; *p* = 0.003) were the variables independently associated with PPM.

The final score obtained at multivariable analysis along with corresponding nomogram are displayed in Figure 6 and Figure 7. The nomogram yielded an AUC of 0.916 (95% CI: 0.850–0.981) for the diagnosis of PPM vs. PDAC.

## 4. Discussion

In the present work, we have described the CT presentation of PPM in 34 patients. The diagnosis of PPM can be suggested by a variety of CT features, which are more frequently seen in patients with PPM than in those with PDAC. Of these, well defined tumor margins (*p* < 0.001), hyperattenuating tumor on arterial phase (*p* < 0.001), maximal tumor enhancement on arterial phase (*p* < 0.001), absence of Wirsung duct enlargement (*p* < 0.001), absence of upstream pancreatic atrophy (*p* < 0.001), absence of vascular involvement (*p* = 0.001) and absence of bile duct dilatation (*p* < 0.001) were the most discriminating features for the diagnosis of PPM against PDAC. It can be assumed that knowledge of discriminating findings and the use of CT-derived nomogram may help clinicians and radiologists favor the diagnosis of PPM in patients with pancreatic mass. This nomogram is based on simple items and does not require any specific CT protocol for data acquisition or expertise as shown by substantial to perfect interobserver agreement between one junior and one senior radiologist.

We observed that hyperattenuating pancreatic mass on arterial phase was more frequently observed in PPM compared to PDAC. In general, PDAC is a poorly vascularized tumor, and hypoattenuating or isoattenuating on all imaging phases (unenhanced, arterial, portal and delayed phases) on CT [30,31]. Regarding PPM, tumor vascularity may depend on the primary tumor but also on the specific chemotherapeutic drug given to the patient. PPMs from RCC and skin melanoma are typically hypervascular, similar to the primary cancer [17,20,22], although PPM from skin melanoma may be hypovascular [32] and RCC may not show typical hyperenhancement [33]. However, this hyperenhancing pattern is observed in up to 75% of PPM and not only in PPM from RCC [17,34].

We found that well-defined margins were a highly discriminating variable to differentiate PPM from PDAC, yielding 74% accuracy for the diagnosis of PPM against PDAC. In the Tsitoutridis et al. study, well defined tumor margins were reported in 9/11 patients (82%) with PPM [17], in 60/79 PPMs (75.8%) in the Klein et al. study [34] and 33/36 PPMs (92%) in the Shi et al. study [25]. By contrast, PDAC more often presents with ill-defined margins [35]. However, this feature conveys some degrees of subjectivity with, as found in our study, the lowest Kappa value (0.717) by comparison with the other qualitative CT variables.

In our study, the absence of bile duct dilatation was another discriminating variable to differentiate PPM from PDAC. Only 4/34 patients (12%) with PPM had bile duct dilatation compared to 19/34 patients (56%) with PDAC. Shi et al. reported bile duct dilatation in only 1/18 patients (6%) with PPM [25]. The low prevalence of bile duct dilatation in PPM may be because it does not originate from the ductal epithelium. Of note, PPM location cannot be considered as a confounding factor because tumors were matched for location and 18/34 (53%) tumors were located in the pancreatic head in both groups.

It has been reported that PPM can invade the Wirsung duct, thus causing Wirsung duct enlargement [22,36]. One series reported obstruction of the Wirsung duct in 25/66 patients (37.9%) with PPM [34], with PPM located in the pancreatic head for 50% of them [34]. By contrast, in the Tsitouridis et al. study, dilatation of Wirsung duct was observed in only 1/11 patients with PPM; this patient had moderate upstream duct dilatation due to PPM from lung adenocarcinoma [17]. Similarly, Shi et al. reported Wirsung duct dilatation in only 2/18 patients with PPM (11%) [25]. In our study, dilatation of the Wirsung duct was observed in only 5/34 patients (15%). Wirsung duct dilatation can cause upstream pancreatic atrophy and this finding was observed in 5/34 patients (5%), and significantly less frequently than in the control group with PDAC.

The absence of visible lymph nodes on CT was another discriminating feature to differentiate PPM from PDAC. Visible lymph nodes were present in only 10/34 patients (29%) with PPM compared to 20/34 patients (65%) with PDAC. This finding has not received particular attention in prior studies making comparison not possible, although researchers have suggested that peripancreatic lymphadenopathy is in favor of PDAC against PPM [37].

In our study, no differences in largest tumor diameter were found between PPM (35.0 ± 21.1 mm) and PDAC (32.1 ± 9.2 mm). This is because patients with PPM and those with PDAC were matched for tumor size. In the Tsitoutridis et al. study, mean diameter of PPM was 2.75 mm (range: 12–52 mm) [17] and 32.2 mm (range: 11–81 mm) in the Shi et al. study [25]. To date, it is thus difficult to determine the value of tumor size for discriminating between PPM and PDAC.

In this study, all PPMs presented as a single pancreatic mass in all patients, whereas multiple PPMs in the same patients have been reported [17,25,34,38]. Single mass is the most frequent pattern, observed in 63.6% to 78.8% of PPMs [17,25,34], whereas multiple PPM has been reported in 16.7% to 27.3% of patients in a series of PPMs from various primaries [17,34]. Multiple PPMs are predominantly observed in patients with RCC, with multiple PPMs observed in up to 45% of patients with RCC [38]. One reason for a null prevalence of multiple PPMs in our study is that the initial search was made using pathologic database and presence of multiple pancreatic mass is a highly suggestive feature for the diagnosis of PPM, thus obviating the need for histopathological confirmation. Of note, in our study all PPM were histopathologically confirmed, whereas in some previous studies a less firm standard of reference was used [17].

In selected patients with PPM, surgical resection can be considered as it may be beneficial in terms of survival [23,24,39]. This is particularly true for patients with PPM from RCC, breast carcinoma and colorectal cancer [40,41]. However, even for inoperable patients, it is of importance to distinguish between PPM and PDAC because treatments markedly differ and also because new minimally invasive approaches may become available in the near future [42,43].

Our study has several limitations. First, the retrospective design has induced selection bias. Second, a relatively small number of patients has been included, but PPM is a relatively rare tumor and most studies reporting imaging features of PPM are a small series. Third, our comparison was based on a 1:1 match so that a different match (i.e., 1:2 or 1:3) might have resulted in different figures. Fourth, the distribution of primary cancers might have influenced the construction of the nomogram and its discriminating capabilities. Fifth, we only compared PPM and PDAC, other primary pancreatic cancers such as neuroendocrine tumors were not included and this comparison should warrant further studies. Sixth, the CT protocol did not include a late phase of enhancement during which PDCAs present with increased enhancement due to desmoplastic growth. Finally, we have compared PPM to PDAC, although other pancreatic tumors, such as neuroendocrine tumors, when hypervascular, may mimic PPM [44,45].

## 5. Conclusions

In conclusion, PPM predominantly presents as an enhancing, predominantly oval and purely solid pancreatic mass with well-defined margins and maximal tumor enhancement during the arterial phase that rarely produces Wirsung duct enlargement and pancreatic parenchyma atrophy by comparison with PDAC. Knowledge of these orientating CT features that can be used in daily practice may help clinicians and radiologists favor the diagnosis of PPM in patients with pancreatic mass. Further research, however, is needed to validate our CT-derived nomogram obtained from a limited sample population.

## Figures and Tables

**Figure 1 cancers-13-03103-f001:**
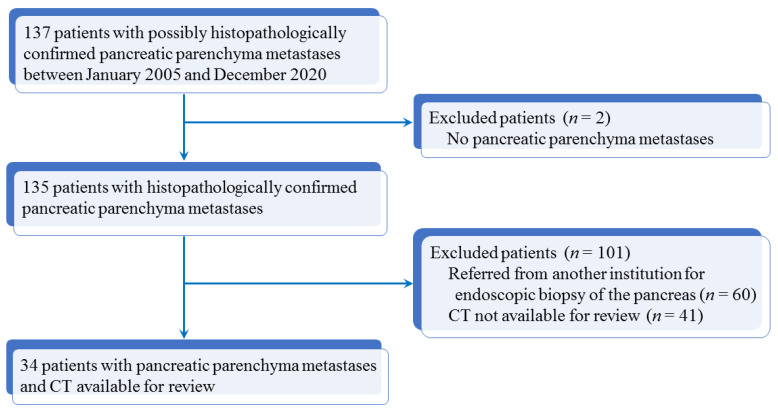
Flowchart shows study inclusion process of patients with pancreatic parenchyma metastases.

**Figure 2 cancers-13-03103-f002:**
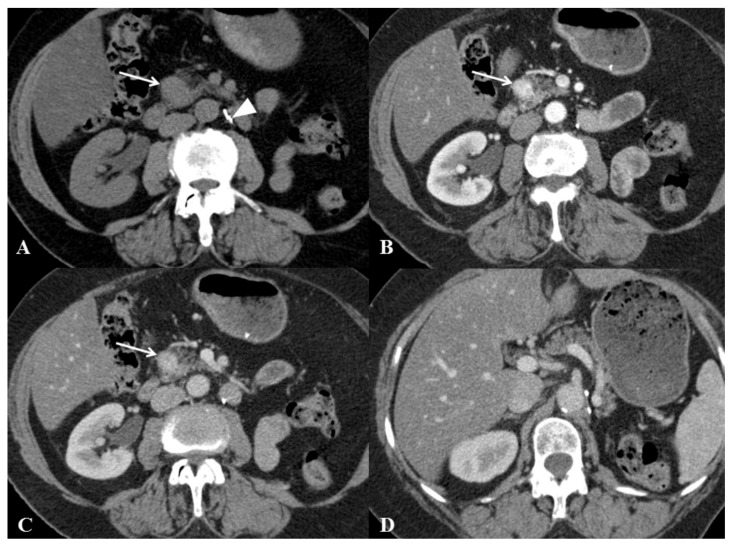
Computed tomography (CT) examination in a 73-year-old woman with prior history of renal cell carcinoma treated by total right nephrectomy four years before. (**A**) Plain CT image of the abdomen in the axial plane shows homogeneous, rounded mass (arrow) of the pancreatic head. No calcifications are present. Metallic clips (arrowhead) indicate prior nephrectomy. (**B**) CT image of the abdomen in the axial plane obtained during the arterial phase of enhancement shows homogeneous, enhancing, purely solid tumor (arrow) of the pancreatic head. (**C**) CT image of the abdomen in the axial plane obtained during the portal venous phase of enhancement shows homogeneous, purely solid tumor (arrow) of the pancreatic head. (**D**) At upper level, CT image shows no Wirsung duct enlargement and no intrahepatic bile duct dilatation. Endoscopic ultrasound-guided biopsy of the pancreatic mass revealed metastasis from renal cell carcinoma. After exclusion of local recurrence and other metastatic sites, the patient underwent pancreaticoduodenectomy (Whipple procedure).

**Figure 3 cancers-13-03103-f003:**
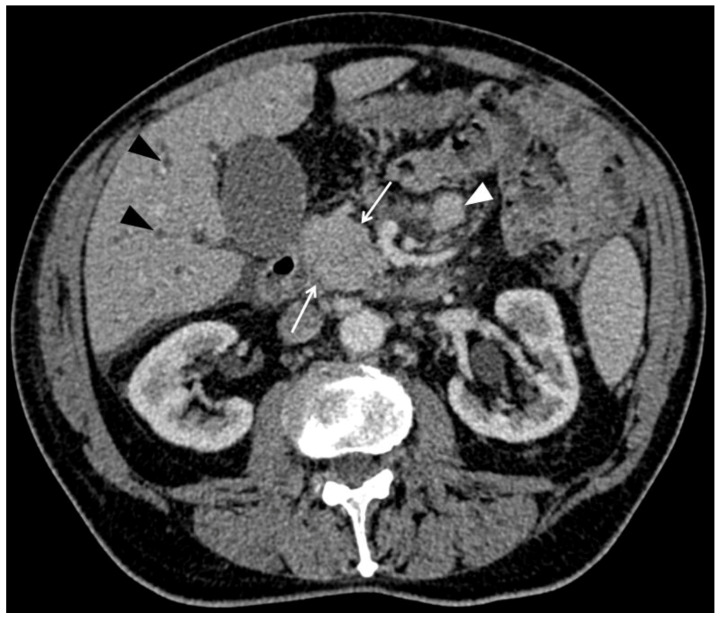
Computed tomography (CT) examination in a 73-year-old man with history of lung adenocarcinoma treated by surgery three years before. The patient was referred for progressive jaundice. No local recurrence of primary lung cancer was found. CT image of the abdomen in the axial plane obtained during the portal venous phase of enhancement shows homogeneous, purely solid mass (arrows) of the pancreatic head in association with intrahepatic bile duct dilatation (black arrowheads), ascites and enlarged mesenteric lymph node (white arrowhead). Endoscopic ultrasound-guided biopsy of the pancreatic mass revealed metastasis from lung adenocarcinoma.

**Figure 4 cancers-13-03103-f004:**
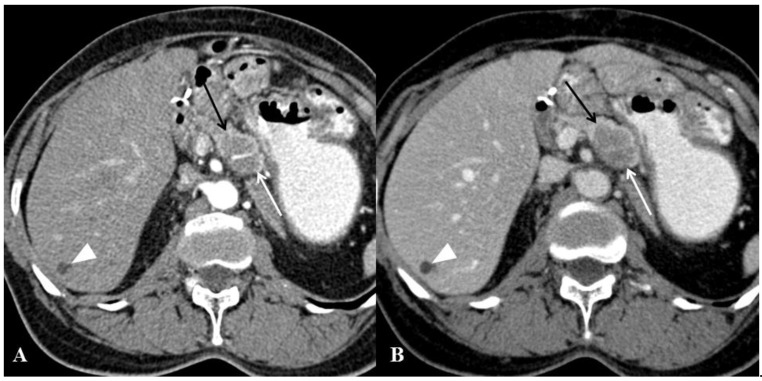
Computed tomography (CT) examination in a 64-year-old woman with prior history of gastric antral adenocarcinoma (pT3N2) treated by surgery and neoadjuvant radiochemotherapy three years before. (**A**,**B**) CT images of the abdomen in the axial plane obtained during the arterial (**A**) and portal venous (**B**) phases of enhancement show well-defined mass (arrows) with heterogeneous content of the pancreatic head. Hepatic lesion (arrowhead) was confirmed as benign biliary cyst and was already present on prior CT examinations. Endoscopic ultrasound-guided biopsy of the pancreatic mass revealed metastasis from gastric adenocarcinoma. After exclusion of local recurrence and other metastatic sites, the patient underwent left pancreatectomy with splenectomy.

**Figure 5 cancers-13-03103-f005:**
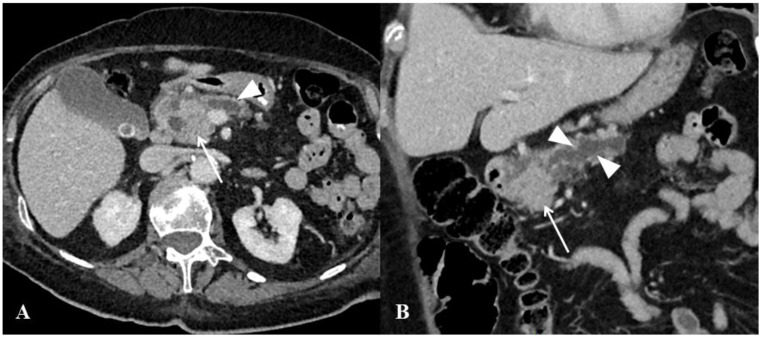
Computed tomography (CT) examination in a 72-year-old woman with pancreatic ductal adenocarcinoma. (**A**,**B**) CT images in the axial (**A**) and coronal (**B**) planes obtained during the portal venous phase show ill-defined mass of pancreatic head that results in upstream Wirsung duct dilatation (arrowheads).

**Figure 6 cancers-13-03103-f006:**
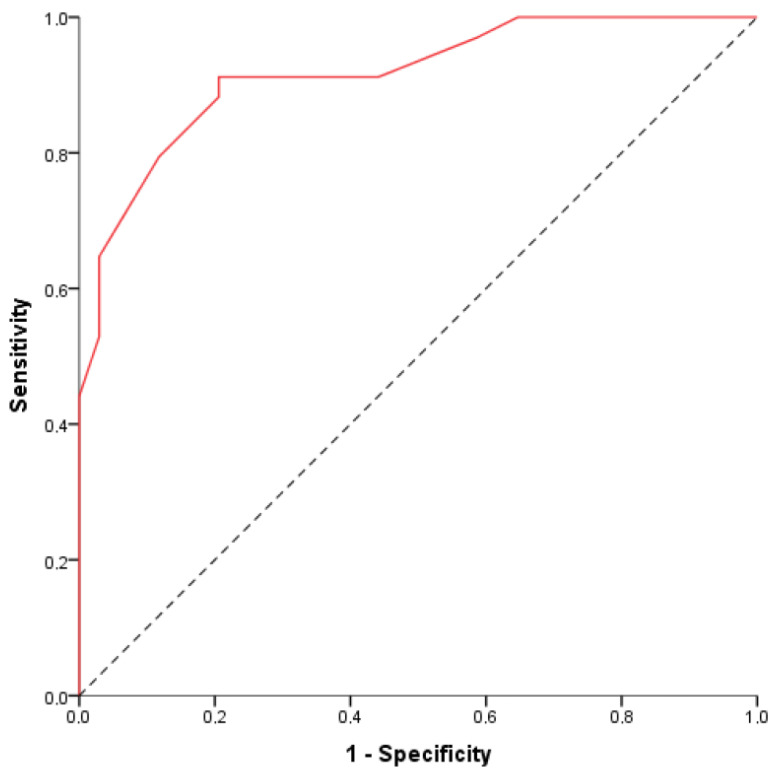
Graph shows area under the curve (AUC) for the final model using the following equation: (1.9 × Well-defined margins) + (1.8 × Maximal enhancement during arterial phase) + (Two Vessel involvement) + (2.4 × No Wirsung duct dilatation) − 4. The final score is obtained using 1 for present and 0 for absent. The area AUC is 0.916 (95% CI: 0.850–0.981).

**Figure 7 cancers-13-03103-f007:**
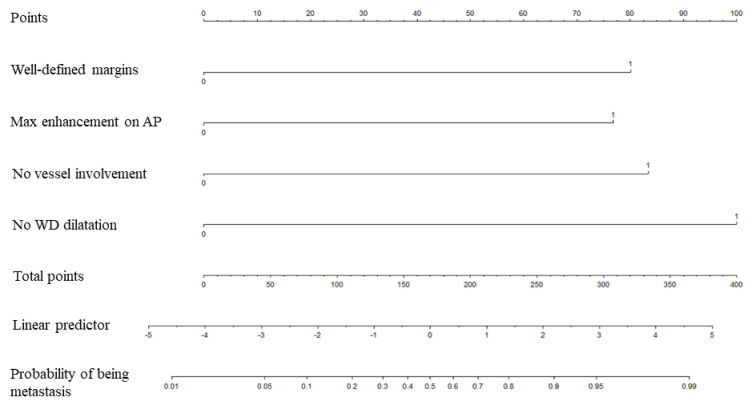
Nomogram for the diagnosis of pancreatic parenchyma metastases vs. pancreatic ductal adenocarcinoma using computed tomography (CT). The CT nomogram was developed using qualitative CT variables. AP indicates arterial phase. WD indicates Wirsung duct.

**Table 1 cancers-13-03103-t001:** Demographics of 34 patients with metastasis to the pancreas and 34 patients with pancreatic ductal adenocarcinoma.

Variable	All Patients (*n* = 68)	PPM Group (*n* = 34)	PDAC Group (*n* = 34)	*p* Value
Age (years)				0.995 *
Mean ± SD	63.5 ± 9.5	63.3 ± 10.2	63.6 ± 8.9	
(median; Q1, Q3)	(64; 57, 70)	(64; 57, 70)	(64.5; 57, 71)	
(range)	(40–82)	(40–82)	(40–82)	
Gender				1 ^‡^
Men	44 (44/68; 65%)	22 (22/34; 65%)	22 (22/34; 65%)	
Women	24 (24/68; 35%)	12 (12/34; 35%)	12 (12/34; 35%)	
Histopathological diagnosis				0.609 ^‡^
Percutaneous biopsy	45 (45/68; 66%)	24 (24/34; 71%)	21 (21/34; 62%)	
Surgical biopsy	23 (23/68; 34%)	10 (10/34; 29%)	13 (13/34; 38%)	
Surgical resection	(23/68; 34%)	10 (11/34; 29%)	13 (13/34; 38%)	0.609 ^‡^

Note. PPM: Pancreatic parenchyma metastases; PDAC: Pancreatic ductal adenocarcinoma. SD indicates standard deviation. Q1 indicates first quartile. Q3 indicates third quartile. * Mann–Whitney U test, ^‡^ Fisher exact test.

**Table 2 cancers-13-03103-t002:** Sensitivity, specificity and accuracy of categorical variables for the diagnosis of pancreatic parenchyma metastasis in 68 patients.

Variable	TP	FP	FN	TN	Sensitivity (%)	Specificity (%)	Accuracy (%)
Tumor shape (round)	15	9	19	25	44 (15/34) (27–62)	74 (25/34) (56–87)	59 (40/68) (46–71)
Well-defined tumor margins	23	7	11	27	68 (23/34) (49–83)	79 (27/34) (62–91)	74 (50/68) (61–84)
Tumor capsule	4	2	30	32	12 (4/34) (3–27)	94 (32/34) (80–99)	53 (36/68) (40–65)
Purely solid content	27	24	7	10	79 (27/34) (62–91)	29 (10/34) (15–47)	54 (37/68) (42–67)
No internal necrosis/hemorrhage	14	17	20	17	41 (14/34) (25–59)	50 (17/34) (32–68)	46 (31/68) (33–58)
Tumor enhancement	34	34	0	0	100 (34/34) (90–100)	68 (23/34) (49–83)	50 (34/68) (38–62)
Homogeneous tumor enhancement	19	11	15	23	56 (19/34) (38–73)	68 (23/34) (49–83)	62 (42/68) (49–73)
Hyperattenuating tumor on arterial phase	11	0	23	34	32 (11/34) (17–51)	100 (34/34) (90–100)	66 (45/68) (54–77)
Maximal tumor enhancement on arterial phase	19	4	15	30	56 (19/34) (38–73)	88 (30/34) (73–97)	72 (49/68) (60–82)
Hypo/isoattenuating tumor on portal phase	34	34	0	0	100 (34/34) (90–100)	0 (0/34) (0–100)	50 (34/68) (38–62)
Tumor calcification	1	1	33	33	3 (1/34) (0–15)	97 (33/34) (85–100)	50 (34/68) (38–62)
No Wirsung duct enlargement	29	11	5	23	85 (29/34) (69–95)	68 (23/34) (49–83)	76 (52/68) (64–86)
No upstream pancreatic atrophy	29	15	5	19	85 (29/34) (69–95)	56 (19/34) (38–73)	71 (48/68) (58–81)
No vascular involvement	27	13	7	21	79 (27/34) (62–91)	62 (21/34) (44–78)	71 (48/68) (58–81)
Segmental portal hypertension	4	9	30	25	12 (4/34) (3–27)	74 (25/34) (56–87)	43 (29/68) (31–55)
Hepatic metastases	2	8	32	26	6 (2/34) (1–20)	76 (26/34) (59–89)	41 (28/68) (29–54)
No bile duct dilatation	30	15	4	19	88 (30/34) (73–97)	56 (19/34) (38–73)	72 (49/68) (60–82)
No visible lymph nodes	24	12	10	22	71 (24/34) (52–85)	65 (22/34) (46–80)	68 (46/68) (55–78)
Adjacent organ involvement	2	3	32	31	6 (2/34) (1–20)	91 (31/34) (76–98)	49 (33/68) (36–61)
Mesenteric panniculitis	3	5	31	29	9 (3/34) (2–24)	85 (29/34) (69–95)	47 (32/68) (35–60)
Ascites	2	5	32	29	6 (2/34) (1–20)	85 (29/34) (69–95)	46 (31/68) (33–58)

Note. TP = true positive. FP = false positive. FN = false negative. TN = true negative. Se = sensitivity (TP/TP + FN). Sp = specificity (TN/TN + FP). Ac = accuracy (TP + TN/TP + FP + TN + FN). Numbers in parentheses are proportions used to calculate the percentages. Numbers in parentheses are exact 95% confidence intervals. All percentages were rounded with no decimals.

**Table 3 cancers-13-03103-t003:** Comparison of CT imaging findings between 34 patients with pancreatic parenchyma metastasis and 34 patients with pancreatic ductal adenocarcinoma.

Variable	PPM (*n* = 34)	PDAC (*n* = 34)	*p* Value
Quantitative variables			
Largest tumor diameter (mm)	35.0 ± 21.1 (13–110) (27.5; 21, 39)	32.1 ± 9.2 (16–59) (32.5; 26, 38)	0.725 *
Visible lymph node size (mm)	20.5 ± 15.3 (9–60) (15; 11, 20)	10.3 ± 3.3 (4–18) (10; 7, 13)	0.014 *
Qualitative variables			
Tumor location Head Body Tail	18 (18/34; 53%) 12 (12/34; 35%) 4 (4/34; 12%)	18 (18/34; 53%) 12 (12/34; 35%) 4 (4/34; 12%)	>0.999 ^†^
Tumor shape Oval Round	19 (19/34; 56%) 15 (15/34; 44%)	25 (25/34; 74%) 9 (9/34; 26%)	0.204 ^‡^
Well-defined tumor margins	23 (23/34; 68%)	7 (7/34; 21%)	<0.001
Tumor capsule	4 (4/34; 12%)	2 (2/34; 6%)	0.673 ^‡^
Purely solid content	27 (27/34; 79%)	24 (24/34; 71%)	0.576 ^‡^
Internal necrosis	14 (14/34; 41%)	17 (17/34; 50%)	0.627 ^‡^
Tumor enhancement	34 (34/34; 100%)	34 (34/34; 100%)	>0.999 ^‡^
Homogeneous tumor enhancement	19 (19/34; 56%)	11 (11/34; 32%)	0.087 ^‡^
Hyperattenuating tumor on arterial phase	11 (11/34; 32%)	0 (0/34; 0%)	<0.001 ^‡^
Maximal tumor enhancement on arterial phase	19 (19/34; 56%)	4 (4/34; 12%)	<0.001 ^‡^
Hypo/isoattenuating tumor on portal phase	34 (34/34; 100%)	34 (34/34; 100%)	>0.999 ^‡^
Tumor calcification	1 (1/34; 3%)	1 (1/34; 3%)	>0.999 ^‡^
No Wirsung duct enlargement	29 (29/34; 85%)	11 (11/34; 32%)	<0.001 ^‡^
No upstream pancreatic atrophy	29 (29/34; 85%)	15 (15/34; 44%)	0.001 ^‡^
No vascular involvement	27 (27/34; 79%)	13 (13/34; 38%)	0.001 ^‡^
Segmental portal hypertension	4 (4/34; 12%)	9 (9/34; 26%)	0.217 ^‡^
Hepatic metastases	2 (2/34; 6%)	8 (8/34; 24%)	0.083
No bile duct dilatation	30 (30/34; 88%)	15 (15/34; 44%)	<0.001 ^‡^
No visible lymph nodes	24 (24/34; 71%)	12 (12/34; 35%)	0.002 ^‡^
Direct adjacent organ involvement	2 (2/34; 6%)	3 (3/34; 9%)	>0.999 ^‡^
Mesenteric panniculitis	3 (3/34; 9%)	5 (5/34; 15%)	0.709 ^‡^
Ascites	2 (2/34; 6%)	5 (5/34; 15%)	0.427 ^‡^

Note. Qualitative variables are expressed as raw numbers; numbers in parentheses are proportions, followed by percentages. Quantitative variables are expressed as mean ± standard deviation (SD) followed by ranges in parentheses; numbers in parentheses are median followed by first (Q1) and third (Q3) quartiles. Bold indicates significant differences. * Mann–Whitney U test; ^‡^ Fisher exact test; ^†^ Freeman–Halton extension of Fisher exact test. No Wirsung duct enlargement corresponds to a Wirsung duct diameter ≤4 mm. PPM: Pancreatic parenchyma metastasis; PDAC: Pancreatic ductal adenocarcinoma.

**Table 4 cancers-13-03103-t004:** Results of univariable analysis with a conditional logistic regression model for 68 patients.

Effect *	Results OR [95% CI]	*p* Value
Round shape	0.46 (0.17–1.26)	0.102
Well-defined tumor margins	8.07 (2.69–24.20)	< 0.001
Tumor capsule	2.13 (0.36–12.51)	0.336
Purely solid content	1.61 (0.53–4.88)	0.288
Internal necrosis	0.70 (0.27–1.83)	0.313
Tumor enhancement	100 (34/34) vs. 100 (34/34) ^†^	> 0.999 ^‡^
Homogeneous tumor enhancement	2.65 (0.99–7.11]	0.043
Hyperattenuating tumor on arterial phase	32 (11/34) vs. 0 (0/34) ^†^	< 0.001 ^‡^
Maximal tumor enhancement on arterial phase	9.50 (2.74–32.95)	< 0.001
Hypo/isoattenuating tumor on portal phase	100 (34/34) vs. 100 (34/34) ^†^	> 0.999 ^‡^
Tumor calcification	1.00 (0.06–16.67)	0.754
No Wirsung duct duct enlargement	12.13 (3.69–39.88)	< 0.001
No upstream pancreatic atrophy	7.35 (2.29–23.57)	< 0.001
No vascular involvement	6.23 (2.11–18.37)	0.001
Segmental portal hypertension	0.37 (0.10–1.35)	0.108
Hepatic metastases	0.20 (0.04–1.04)	0.042
No bile duct dilatation	9.50 (2.739–32.95)	< 0.001
No visible lymph nodes	4.80 (1.75–13.21)	0.002
Direct adjacent organ involvement	0.65 (0.10–4.13)	0.500
Mesenteric panniculitis	0.56 (0.12–2.56)	0.355
Ascites	0.36 (0.07–2.01)	0.214

Note. Unless otherwise noted, data are odds ratios, 95% exact confidence intervals are in parentheses. Odds ratio and 95% CIs are not shown for some variables because a zero value for corresponding data in Table 3 led to unstable estimates of these parameters. * All effects are present vs. absent. Wirsung duct enlargement corresponds to a Wirsung duct diameter > 4 mm. ^†^ Frequency of corresponding variable; data are percentages; proportions are in parentheses. ^‡^ Exact conditional logistic regression.

## Data Availability

The data presented in this study are available on request from the corresponding author.

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
