# Peer review of "Utility of CT to Differentiate Pancreatic Parenchymal Metastasis from Pancreatic Ductal Adenocarcinoma"

_cancers, 2021, doi:10.3390/cancers13133103_

Round 1
Reviewer 1 Report
I appreciate the opportunity to review this work. I have a few comments for the authors to consider.
1) Sample size is very small. This is understandable given the infrequent incidence of PPM. Can these conclusions be useful as a generalizable tool, given this limitation, or is this nomogram difficult to apply in a real world setting?
2) What about other primary pancreatic cancers that are not adenocarcinoma, such as a PNET? Would these fall in line with PDAC or PPM based upon your work?
3) I suspect that most practitioners, finding a tumor in the pancreas in a patient with other metastatic disease, would still biopsy the pancreatic lesion. In what instances would your nomogram eliminate the need for such a biopsy?
4) Is there any data on how frequently a cancer mets to the pancreas? You cite the cancers that are most common to do so, but within those diseases, how common is that to happen and should practitioners be more expectant of this finding for those tumor types?
5) What impact will the skill of the radiologist have in interpreting scans using this approach?
Author Response
Reviewer #1
I appreciate the opportunity to review this work. I have a few comments for the authors to consider.
- Sample size is very small. This is understandable given the infrequent incidence of PPM. Can these conclusions be useful as a generalizable tool, given this limitation, or is this nomogram difficult to apply in a real world setting?
Thank you for this comment. The nomogram is based on simple visual items on CT and does not require any specific CT acquisition protocol nor specific expertise in imaging as shown by as shown by substantial to perfect interobserver agreement between one junior and one senior radiologists for the variables used to build the nomogram. It can be used in daily practice.
We added a sentence in the discussion as follows: “This nomogram is based on simple items and does not require any specific CT protocol for data acquisition or expertise as shown by substantial to perfect interobserver agreement between one junior and one senior radiologists.
- What about other primary pancreatic cancers that are not adenocarcinoma, such as a PNET? Would these fall in line with PDAC or PPM based upon your work?
Thank you for this very interesting comment. You are totally right: It would be very interesting to compare other primary pancreatic tumors with pancreatic metastases. However PNET as PPM are rare conditions and the power of a such study would be limited without a large cohort. So we decided to focus on pancreatic adenocarcinomas which are more common primary pancreatic tumors. With your approval, we added a sentence in the limitation paragraph of the discussion as follows: “Fifth, we only compared PPM and PDAC, other primary pancreatic cancer as neuroendocrine tumors were not included, and this comparison should warrant further studies. »
- I suspect that most practitioners, finding a tumor in the pancreas in a patient with other metastatic disease, would still biopsy the pancreatic lesion. In what instances would your nomogram eliminate the need for such a biopsy?
Thank you for this comment. Pancreatic biopsies convey some technical difficulties, require in most of cases an echo-endoscopy under general anesthesia and are associated with a non negligeable risk of acute pancreatitis. The number of patients and the retrospective design of our study do not allow us to conclude that pancreatic biopsies are no more required for the diagnosis; however, it could be an interesting topic for future prospective work in terms of patient management.
- Is there any data on how frequently a cancer mets to the pancreas? You cite the cancers that are most common to do so, but within those diseases, how common is that to happen and should practitioners be more expectant of this finding for those tumor types?
Thank you for this comment. We describe these epidemiologic points in the second paragraph of the introduction as follow: “Pancreatic parenchyma metastasis (PPM) is a rare tumor that represents 2-5% of all malignant tumors of the pancreas [9]. PPM is a rare condition that is found in 3-12% of patients with advanced cancer in autopsy studies [10; 11] »
- What impact will the skill of the radiologist have in interpreting scans using this approach?
Thank you for this comment. As found in our study, the CT findings used to discriminate between PDAC and PPM convey a substantial to perfect interobserver agreement between one junior and one senior radiologists, so that the skill of the radiologist has limited influence of image analysis.
Reviewer 2 Report
In current study, the authors radiographically characterize matched cases of PPM and de novo PDAC to provide features that can be used in discriminating the two. This analyses reveals numerous radiographic characteristics associated with one, but not the other ultimately resulting in a nomogram to guide providers. The study is interesting and uses a sound approach to address a rare to significant clinical problem. I have several comments for the authors.
1) The simple summary in its current form would be difficult for a lay person to understand.
2) Throughout the manuscript, the authors report to the thousandth decimal position, resulting anywhere between 3-5 significant figures. It is uncommon to report this level of certainty and at times it can make certain sections difficult to read
3) There are numerous typos in the introduction, such as line 44. Please read through carefully and address
4) Figure legends for figures 2 and 3 are provided for the wrong figures
5) In its current form, the nomogram is very small and difficult to read.
6) As the authors indicate that clinical history alone cannot help distinguish between PPM and PDAC, what percentage of PDAC patients had a history of prior malignancy in this study?
Author Response
Reviewer #2
In current study, the authors radiographically characterize matched cases of PPM and de novo PDAC to provide features that can be used in discriminating the two. This analysis reveals numerous radiographic characteristics associated with one, but not the other ultimately resulting in a nomogram to guide providers. The study is interesting and uses a sound approach to address a rare to significant clinical problem. I have several comments for the authors.
1) The simple summary in its current form would be difficult for a lay person to understand.
Thank you for this comment. The simple summary has been updated and is now more straightforward.
2) Throughout the manuscript, the authors report to the thousandth decimal position, resulting anywhere between 3-5 significant figures. It is uncommon to report this level of certainty and at times it can make certain sections difficult to read
Thank you for this comment. We changed all quantitative values from 0.001 to 0.01.
3) There are numerous typos in the introduction, such as line 44. Please read through carefully and address
This you for this careful reading. A careful checking has been made and two grammatical errors have been found and fixed accordingly.
4) Figure legends for figures 2 and 3 are provided for the wrong figures
Thank you for this comment. There was a shift between the two legends. Change has been made accordingly.
5) In its current form, the nomogram is very small and difficult to read.
Thank you for this comment. The nomogram is now a full figure and larger.
6) As the authors indicate that clinical history alone cannot help distinguish between PPM and PDAC, what percentage of PDAC patients had a history of prior malignancy in this study?
Thank you for this comment. This has been added in the reviewed manuscript in the M&M section Four patients had prior history of another cancer including breast carcinoma, lung carcinoma, colon adenocarcinoma and skin melanoma (one patient each).
Reviewer 3 Report
Dear authors,
this is a well-written manuscript addressing a common clinical topic which we all are confronting with in the daily routine.
The discriminatory features the authors describe here are not really new and I guess every radiologist dealing with oncologic imaging has made similar experiences in its daily practice.
Due to the relatively small number of patients evaluated, significances of all these features have to be questioned.
Nevertheless, the authors have done a good job and deserve therefore our compliments.
An interesting point is not been addressed in this stdy, presumably due to its retrospective character. Knowingly, PDCAs present with desmoplastic growth and increased enhancement in the equilibrium phase contrary to most metastases.
Author Response
Reviewer #3
Dear authors,
- This is a well-written manuscript addressing a common clinical topic which we all are confronting with in the daily routine.
The discriminatory features the authors describe here are not really new and I guess every radiologist dealing with oncologic imaging has made similar experiences in its daily practice.
- Due to the relatively small number of patients evaluated, significances of all these features have to be questioned.
Thank you very much, this is correct and our conclusion is in line with these concerns. These results have to be confirmed prospectively as stated in the Discussion section.
- Nevertheless, the authors have done a good job and deserve therefore our compliments.
Thank you for this comment.
- An interesting point is not been addressed in this study, presumably due to its retrospective character. Knowingly, PDCAs present with desmoplastic growth and increased enhancement in the equilibrium phase contrary to most metastases.
Thank you for this comment. Unfortunately, late phase CT is not commonly performed. After intravenous administration of iodinated contrast material, all patients underwent arterial and portal phase images and analysis of late enhancement was not possible. But this point should deserve further investigation. This comment has been addressed in the limitation section of the revised manuscript as follows: Sixth, the CT protocol did not include a late phase of enhancement during which PDCAs present with increased enhancement due to desmoplastic growth
Round 2
Reviewer 1 Report
Thank you for considering my comments
Reviewer 3 Report
Thanks to the authors for the manuscript improvement.